# Energy diffusion and the butterfly effect in inhomogeneous Sachdev-Ye-Kitaev chains

**Yingfei Gu, Andrew Lucas and Xiao-Liang Qi**

Department of Physics, Stanford University, Stanford, CA 94305, USA

yfgu@stanford.edu, ajlucas@stanford.edu, xlqi@stanford.edu

## Abstract

We compute the energy diffusion constant $D$, Lyapunov time $\tau_{\mathrm{L}}$ and butterfly velocity $v_{\mathrm{B}}$ in an inhomogeneous chain of coupled Majorana Sachdev-Ye-Kitaev (SYK) models in the large $N$ and strong coupling limit. We find $D \le v_{\mathrm{B}}^2 \tau_{\mathrm{L}}$ from a combination of analytical and numerical approaches. Our example necessitates the sharpening of postulated transport bounds based on quantum chaos.

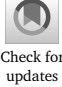 Check for updates

## 1 Introduction

A few years ago it was noted that many experimentally realized "strange metals" seem to be characterized by Drude "transport time" of order $\tau_* \sim \hbar/k_{\mathrm{B}}T$ [1]. As this time scale was proposed to be the "fastest possible" time scale governing quantum dynamics [2, 3], it was conjectured by Hartnoll [4] that these strongly interacting strange metals remained metallic

due to a fundamental bound on diffusion: $D \gtrsim v^2 \tau_*$. In strongly interacting systems without quasiparticles, many-body quantum chaos provides a natural velocity $v$ and time scale $\tau_*$ for such a diffusion bound [5,6]. In some large-$N$ models, it is natural to define a Lyapunov time $\tau_L$ and butterfly velocity $v_B$ as [7,8]

$$\langle V(x,t)W(0,0)V(x,t)W(0,0)\rangle_T \sim 1 - \frac{1}{N}\exp\left[\frac{t}{\tau_L} - \frac{x}{\tau_L v_B}\right], \qquad (1)$$

for rather general Hermitian operators $V$ and $W$. $\tau_L$ is an analogue of the Lyapunov time from classical chaos: it describes the rate at which quantum coherence is lost. Similarly, $v_B$ is called the butterfly velocity: it governs the speed of chaos propagation, and is somewhat analogous to a state-dependent Lieb-Robinson velocity [9]. Since we now have a velocity and time scale which can be defined in any strange metal, [5,6] noted that $D$ was naturally related to $v_B^2 \tau_L$ in some simple holographic settings. A natural question to ask is whether Hartnoll's conjecture becomes

$$D \gtrsim v_B^2 \tau_L. \qquad (2)$$

Originally (2) was observed to hold for charge diffusion constant [5], with theory-dependent O(1) prefactors, but there are now multiple known counterexamples [10–12]. It is more compelling that (2) hold for the energy diffusion constant: as argued in [11,13,14], $v_B$ characterizes the loss of quantum coherence, a process related to quantum "phase relaxation" which should also characterize energy fluctuations and diffusion. Furthermore, additional evidence for an energy diffusion bound of the form (2) has arisen in holographic models in the low temperature limit [6,15], and models of Fermi surfaces coupled to gauge fields [13]: at weak coupling, [16,17] have also proposed a relation between diffusion and chaos.

Much of the recent literature focuses on "homogeneous" models of disorder: these can crudely be thought of as models where momentum is not a conserved quantity (hence, by Noether's Theorem, microscopic translation symmetry has been broken), yet the effective equations governing transport remain spatially homogeneous. However, transport coefficients can be sensitive to how translation symmetry has been broken [14], so it is important to test the robustness of any transport bound in inhomogeneous models. Such a test was performed for charge diffusion in a family of holographic models [10], and the inequality in (2) was found to be reversed. In this paper, we test (2) for the energy diffusion constant in an inhomogeneous system: an inhomogeneous analogue of a Sachdev-Ye-Kitaev (SYK) chain of Majorana fermions. We will describe this solvable model of a "strange metal" without quasiparticles in more detail in Section 2. Our main result is that in this model the energy diffusion constant is *upper bounded* by chaos:

$$D \lesssim v_B^2 \tau_L. \qquad (3)$$

We will prove this in the limit where the inhomogeneity is parametrically slowly varying, and provide examples where the ratio $D/v_B^2 \tau_L$ is arbitrarily small, in Section 3.

Thus, we do not expect (2) to be generically true in disordered strange metals: holding as a strict inequality up to a finite O(1) prefactor which may be theory-dependent[1] but robust to disorder. Furthermore, in a generic, nearly translation invariant $1+1$ dimensional field theory without a global U(1) symmetry, the natural diffusion constant is the diffusion of energy. As noted in [6], this diffusion constant will be parametrically large due to the fact that translations are only weakly broken [14]. Hence, we now have examples of strange metals where $D$ is either much larger or much smaller than $v_B^2 \tau_L$. It is still an interesting open question whether transport properties of strange metals are related to quantum chaos. Any such relation may

---

[1]For example, [13] found $D \approx 0.42 v_B^2 \tau_L$ in their model, in a clean theory. In the SYK chain that we study, the numerical prefactor is 1, and in SYK-like holographic models the constant ranges from 0.5 to 1 [15]. Examples where the prefactor can be arbitrarily small can be found in [6].

be restricted to particular diffusion constants and/or models. For example, it may be the case that diffusion constants of translation invariant field theories are related to chaos [13,15]. We hope that variational methods, related to those developed in [18–20] for hydrodynamic and holographic models, may be useful in providing rigorous bounds on transport and chaos in disordered strange metals.

For the remainder of the paper, we set $\hbar = k_{\mathrm{B}} = 1$.

## 2 The Inhomogeneous SYK Chain

The SYK model is a strongly interacting large-$N$ model in $0+1$ spacetime dimensions. It was introduced a long time ago as a model of disordered quantum magnets [21,22]; it was revived more recently [23] due to its possible connection to $\mathrm{AdS}_2$ holography [24,25], which is a toy model for quantum gravity [26–31]. Although it has since been shown that the SYK model does not admit a simple holographic dual [32], it does share many fascinating properties of holographic theories, including being "maximally chaotic" [33].

The model that we introduce is a generalization of the SYK chain model developed in [34].[2] Consider a one-dimensional lattice of $L$ sites, where on each lattice site $x$ there exist $N$ Majorana fermions $\chi_{i,x}$ ($i = 1, \dots, N$), obeying the standard commutation relationship $\{\chi_{i,x}, \chi_{j,y}\} = \delta_{ij}\delta_{xy}$. The Hamiltonian of these fermions is

$$H = \sum_{x=1}^{L}\left( \sum_{i<j<k<l} J_{ijkl,x}\chi_{i,x}\chi_{j,x}\chi_{k,x}\chi_{l,x} + \sum_{i<j,k<l} J'_{ijkl,x}\chi_{i,x}\chi_{j,x}\chi_{k,x+1}\chi_{l,x+1} \right), \qquad (4)$$

where the couplings $\{J_{ijkl,x}\}$ and $\{J'_{ijkl,x}\}$ are all assumed to be independent Gaussian random variables drawn from a distribution with zero mean and following variance:

$$\overline{J_{ijkl,x}J_{i'j'k'l',x}} = \frac{3!}{N^3}J_{0,x}^2 \delta_{ii'}\delta_{jj'}\delta_{kk'}\delta_{ll'}\delta_{xy}, \quad \overline{J'_{ijkl,x}J'_{i'j'k'l',x}} = \frac{1}{N^3}J_{1,x}^2 \delta_{ii'}\delta_{jj'}\delta_{kk'}\delta_{ll'}\delta_{xy}, \quad (5)$$

The important (and only) difference comparing to [34] is that we do not assume the variances $J_{1,x}^2$ and $J_{0,x}^2$ take the same value for each $x$. We are interested in the thermodynamic limit $L \to \infty$.

One can show that the replica-diagonal[3] partition function, at the inverse temperature $\beta = 1/T$, can be written as a path integral over two bilocal fields:

$$\overline{Z} = \int \mathrm{D}G\, \mathrm{D}\Sigma\, \exp\left[-N S_{\mathrm{eff}}[G, \Sigma]\right] \qquad (6)$$

with the Euclidean time action

$$S_{\mathrm{eff.}} = \sum_{x=1}^{L}\left\{ -\log \mathrm{Pf}(\partial_\tau - \Sigma_x) + \frac{1}{2}\int_0^\beta \mathrm{d}^2\tau \left[ \Sigma_x G_x - \frac{J_{0,x}^2}{4}G_x^4 - \frac{J_{1,x}^2}{4}G_x^2 G_{x+1}^2 \right] \right\} \qquad (7)$$

The "Green's function" $\{G_x(\tau_1, \tau_2)\}$ and "self energy" $\{\Sigma_x(\tau_1, \tau_2)\}$ are functions of two time variables. The product $\Sigma_x G_x$ in above formula is abbreviation for $\Sigma_x(\tau_1, \tau_2)G_x(\tau_1, \tau_2)$; $G_x^4$ and $G_x^2 G_{x+1}^2$ are similar products. $J_{0,x}^2$ and $J_{1,x}^2$ show up to exactly quadratic order in (7) because the random couplings $J_{ijkl,x}$ and $J'_{ijkl,x}$ were Gaussian random variables. As the manipulations in this section are essentially identical to [32,34], we only present the few steps

---

[2]See [35,36] for another way of adding a spatial dimension to the SYK model.

[3]Off-diagonal sectors in replica space do not contribute at the orders in $1/N$ that we study.

where they differ in an important way (due to the absence of translational invariance on average).

It is convenient to rewrite the interaction term ($G^4$-term) into the following form:

$$\sum_x \left( J_{0,x}^2 G_x^4 + J_{1,x}^2 G_x^2 G_{x+1}^2 \right)$$
$$= \sum_x \left\{ \left( J_{0,x}^2 + \frac{J_{1,x}^2 + J_{1,x-1}^2}{2} \right) G_x^4 + \frac{1}{2} G_x^2 \left[ J_{1,x}^2 \left( G_{x+1}^2 - G_x^2 \right) + J_{1,x-1}^2 \left( G_{x-1}^2 - G_x^2 \right) \right] \right\}. \quad (8)$$

If one chooses $J_{0,x}$ and $J_{1,x}$ such that for each $x$,

$$J^2 \equiv J_{0,x}^2 + \frac{J_{1,x}^2 + J_{1,x-1}^2}{2} \quad (9)$$

is a constant independent of $x$, then the effective on-site coupling is easily seen to be $x$-independent. The saddle point equations of $S_{\text{eff}}$ become

$$G_x^{-1}(\tau_1, \tau_2) = \Sigma_x(\tau_1, \tau_2) - \delta'(\tau_1 - \tau_2), \quad (10a)$$

$$\Sigma_x(\tau_1, \tau_2) = J_{0,x}^2 G_x(\tau_1, \tau_2)^3 + \frac{J_{1,x}^2 G_{x+1}(\tau_1, \tau_2)^2 + J_{1,x-1}^2 G_{x-1}(\tau_1, \tau_2)^2}{2} G_x(\tau_1, \tau_2), \quad (10b)$$

and they admit an $x$-independent approximate solution: $G_x^s(\tau_1, \tau_2) = G^s(\tau_1 - \tau_2)$, with

$$G^s(\tau) = b^\Delta \left( \frac{\beta J}{\pi} \sin \frac{\pi \tau}{\beta} \right)^{-2\Delta}, \quad 0 \leqslant \tau < \beta \quad (11a)$$

$$b = \frac{1}{\pi} \left( \frac{1}{2} - \Delta \right) \tan(\pi \Delta), \quad \Delta = \frac{1}{4},$$

which becomes exact at $\beta J \to \infty$ (conformal) limit. The system also has a uniform specific heat per site $c \approx \frac{0.396}{\beta J}$. Thus, as in [34], this saddle point is identical to the $0 + 1$-dimensional SYK model of [23, 32] at coupling constant $J$. If the choice (9) is not made, then the saddle point equations do not admit a homogeneous solution, and it is unclear what the effective theory is.

In the strong coupling limit, $N \gg \beta J \gg 1$, and long wavelength limit, the physics of interest to us is governed by the fluctuations induced by reparametrization modes $f_x \in \text{Diff}(S^1)$, which act as $G(\tau_1, \tau_2) \to (f_x'(\tau_1) f_x'(\tau_2))^{1/4} G(f_x(\tau_1), f_x(\tau_2))$. To quadratic order of the infinitesimal fluctuations, and leading order in $1/\beta J$ expansion, the effective action for the fluctuations has a simple form in Fourier space: defining $f_x(\tau) = \tau + \epsilon_x(\tau)$, $\epsilon_n = \int_0^\beta d\tau e^{\frac{2\pi i n \tau}{\beta}} \epsilon(\tau)$, we find

$$S_{\text{eff.}} = \frac{1}{256\pi} \sum_{xy} \sum_n \epsilon_{n,x} |n|(n^2 - 1) \left( \alpha \frac{|n|}{\beta J} \delta_{xy} + C_{xy} \right) \epsilon_{-n,y}, \quad (12)$$

where all $x$-dependence is contained in the tridiagonal matrix

$$C_{xy} = \frac{1}{3J^2} \begin{pmatrix} \ddots & -J_{1,x-1}^2 & 0 & 0 \\ -J_{1,x-1}^2 & J_{1,x-1}^2 + J_{1,x}^2 & -J_{1,x}^2 & 0 \\ 0 & -J_{1,x}^2 & J_{1,x}^2 + J_{1,x+1}^2 & -J_{1,x+1}^2 \\ 0 & 0 & -J_{1,x+1}^2 & \ddots \end{pmatrix}, \quad (13)$$

and $\alpha = \sqrt{2}\alpha_K \approx 12.7$ is a constant determined by numerics [32]. A few more steps of this derivation are contained in Appendix A. The long wavelength limit alluded to earlier is the

regime when the eigenvalues of $C_{xy}$ are not larger than $1/\beta J$, which (as we will see) do exist even for the disordered matrix. The derivation of this effective action is identical to [34]: in this previous work, $C_{xy}$ was translation invariant and so (12) was written in momentum space, where the matrix $C_{xy}$ becomes diagonal.

By writing $C$ in the form

$$
\begin{aligned}
C_{xy} &= D_{xz}^{\mathsf{T}} \Lambda_{zw} D_{wy} \\
&= \frac{1}{3J^2}
\begin{pmatrix}
\ddots & -1 & 0 & 0 \\
0 & 1 & -1 & 0 \\
0 & 0 & 1 & -1 \\
0 & 0 & 0 & \ddots
\end{pmatrix}^{\mathsf{T}}
\begin{pmatrix}
\ddots & 0 & 0 & 0 \\
0 & J_{1,x}^2 & 0 & 0 \\
0 & 0 & J_{1,x+1}^2 & 0 \\
0 & 0 & 0 & \ddots
\end{pmatrix}
\begin{pmatrix}
\ddots & -1 & 0 & 0 \\
0 & 1 & -1 & 0 \\
0 & 0 & 1 & -1 \\
0 & 0 & 0 & \ddots
\end{pmatrix}
\end{aligned}
\tag{14}
$$

we immediately recognize that it is positive definite and can be interpreted as the first-order finite-difference discretized version of the differential operator

$$
C_{xy} \sim \frac{1}{3J^2} \left( -\frac{\mathrm{d}}{\mathrm{d}x} J_1(x)^2 \frac{\mathrm{d}}{\mathrm{d}x} \right)_{\text{discretized}}.
\tag{15}
$$

The interpretation of $C_{xy}$ as an approximate differential operator becomes exact when $J_{1,x}^2$ varies slowly. Letting $\mathbb{E}$ denote spatial averages over $x$, suppose that

$$
\mathbb{E}\left[ J_{1,x}^2 J_{1,y}^2 \right] \sim f\left( \frac{|x-y|}{M} \right)
\tag{16}
$$

with $M$ a (large) integer, and $f(x)$ a non-zero function for O(1) argument. To leading order in $1/M$, the low-lying spectrum of the discrete operator $C_{xy}$ will be identical to the continuum differential operator (15).

In our model, we will take $J_{1,x}$ to be an arbitrary function of $x$, simply constrained to $0 \leqslant J_{1,x}^2 \leqslant J^2$ (otherwise $J_{0,x}^2$ as defined in (9) would be negative). The properties of the matrix $C_{xy}$ will then depend on the inhomogeneity that we encode through $x$-dependent $J_{1,x}$.

## 3 Diffusion and Chaos

From the effective action (12), we are able to extract the thermal response functions. The procedure is identical to [34] and a diffusion pole is found in the energy density ($T^{tt}$) two-point function:

$$
\langle T_{x,n}^{tt} T_{y,-n}^{tt} \rangle_T \sim \left( |\omega_n| \delta_{xy} + \frac{2\pi J}{\alpha} C_{xy} \right)^{-1} \equiv \left( |\omega_n| \delta_{xy} + \widetilde{C}_{xy} \right)^{-1}.
\tag{17}
$$

Upon proper analytic continuation to real time, we interpret (17) as having diffusive poles (on the negative imaginary axis) whenever $\mathrm{i}\omega$ is an eigenvalue of $\widetilde{C}_{xy}$. $\widetilde{C}_{xy}$ is analogous to a tight-binding-model hopping matrix. If $\widetilde{C}_{xy}$ commutes with a discrete translation operator, then we expect plane wave eigenstates, the lowest-lying of which will have an eigenvalue $\sim L^{-2}$ in a chain of length $L$. If $\widetilde{C}_{xy}$ is random, then strictly speaking all eigenstates of $\widetilde{C}_{xy}$ *at fixed $\omega$* are localized in the continuum. However, because $\widetilde{C}_{xy}$ is analogous to a discretized differential operator (15) which has an exact delocalized zero mode, the low-lying spectrum of $\widetilde{C}_{xy}$ will look diffusive on length scales $L$. In other words, the localization length grows faster than $\omega^{-1/2}$ [37], and the smallest nontrivial eigenvalue scales as $L^{-2}$ in this case as well. Hence, the diffusion constant $D$ is finite.

In fact, the lowest-lying non-trivial eigenvectors $u_x$ of $\widetilde{C}_{xy}$ are well approximated by plane waves:

$$u_x \sim \mathrm{e}^{\mathrm{i}qx}, \quad q = \pm\frac{2\pi}{L}, \tag{18}$$

which can be verified numerically. See Appendix B for more comments on this equation. The eigenvalue of such a $u_x$ will be $Dq^2$, with $D$ an effective diffusion constant. In the large $M$ limit, we may compute $D$ by solving the following differential equation as $q \to 0$:

$$-\frac{\mathrm{d}}{\mathrm{d}x}\left(D(x)\frac{\mathrm{d}u}{\mathrm{d}x}\right) = Dq^2 u. \tag{19}$$

The constant $D$ is computed in Appendix B:

$$\frac{1}{D} = \mathbb{E}\left[\frac{1}{D(x)}\right]. \tag{20}$$

This equation has a straightforward physical interpretation. Because the specific heat in our model is $x$-independent to leading order [34], $D$ is proportional to the thermal conductivity. One can approximate our inhomogeneous chain by joining together homogeneous SYK chains of length $L' \ll M$. Within each of these chains, the thermal conductivity is proportional to the diffusion constant $D(x)$. Joining together these segments of length $L'$, we find a resistor network: hence, the thermal resistivity spatially averages. This leads to (20). In Appendix B, we argue that (20) is applicable even beyond the large $M$ limit, under some assumptions which work relatively well in practice numerically, so long as finite size effects are small.

Now we study the butterfly velocity, defined by out-of-time-ordered correlation functions of spatially separated operators. In order to extract $v_{\mathrm{B}}$, we study the (properly regularized) connected out-of-time-ordered correlation function. One finds [34], in the region $\beta \ll t \lesssim \beta \log N$ that

$$\frac{1}{N^2}\sum_{i,j}\left\langle \chi_{i,x}(t), \chi_{j,y}(0)\chi_{i,x}(t), \chi_{j,0}(0)\right\rangle_{T,\mathrm{connected}} \sim \mathrm{e}^{\frac{2\pi}{\beta}t}\left(\frac{2\pi}{\beta} + \widetilde{C}_{xy}\right)^{-1}. \tag{21}$$

Comparing to (1), we observe that in this model, as in the usual SYK model,

$$\tau_{\mathrm{L}} = \frac{\beta}{2\pi}. \tag{22}$$

This matrix inverse is the discrete analogue of the Green's function

$$\frac{2\pi}{\beta}G(x;y) - \frac{\mathrm{d}}{\mathrm{d}x}\left(D(x)\frac{\mathrm{d}G(x;y)}{\mathrm{d}x}\right) = \delta(x-y). \tag{23}$$

In the long range disorder limit, when at each point $x$

$$M \gg \sqrt{\frac{2\pi}{\beta D(x)}}, \tag{24}$$

the solution of this equation is exponentially decaying [10]:

$$G(x;y) \sim \mathrm{e}^{-|x-y|/v_{\mathrm{B}}\tau_{\mathrm{L}}}, \tag{25}$$

with

$$\frac{1}{v_{\mathrm{B}}} = \sqrt{\frac{2\pi}{\beta}}\mathbb{E}\left[\frac{1}{\sqrt{D(x)}}\right]. \tag{26}$$

This equation can be derived by noting that, away from the points $x = y$, we can change coordinates from $x$ to $s$, defined by $D(x)\partial_x \equiv \partial_s$. One then finds

$$\frac{2\pi}{\beta}D(s)G = \frac{\mathrm{d}^2 G}{\mathrm{d}s^2}, \tag{27}$$

and when $D(s)$ varies slowly, one can straightforwardly write

$$G = \exp\left[-\int \mathrm{d}s \sqrt{\frac{2\pi D(s)}{\beta}}\right] = \exp\left[-\int \mathrm{d}x \sqrt{\frac{2\pi}{D(x)\beta}}\right]. \tag{28}$$

Hence, we find that $D$ and $v_{\mathrm{B}}$ are not equal. Using the Cauchy-Schwarz inequality it is straightforward to conclude

$$v_{\mathrm{B}}^2 \tau_{\mathrm{L}} = \mathbb{E}\left[\frac{1}{\sqrt{D(x)}}\right]^{-2} \geqslant \mathbb{E}\left[\frac{1}{D(x)}\right]^{-1} = D. \tag{29}$$

The physics at play here is essentially the same as in holography, where charge diffusion was shown to obey a similar inequality, for the same reasons [10].

We have numerically computed $D$ and $v_{\mathrm{B}}$ in SYK chains of finite length $L$ with periodic boundary conditions. $D$ is found by averaging the two smallest non-vanishing eigenvales of $C_{xy}$. $v_{\mathrm{B}}^{-1}$ is found by computing the typical value of $-\log G(x; y)/|x - y|$ for $|x - y| \sim L/2$.[4] So long as $D(x) > D_{\min} > 0$, we find that $D$ agrees with the "resistor chain" prediction (20) for any $M$, so long as $L/M \gtrsim 20$, to within about 0.1% residual error (which is possibly a numerical finite size effect). Indeed, the derivation of (20) in Appendix B does not rely on the assumption that $J_1$ is slowly varying, so this is not surprising. As shown in Figure 1, we see that while $D$ agrees very well with the "hydrodynamic" theory, $v_{\mathrm{B}}^2$ agrees with the hydrodynamic theory at large $M$, while partly approaching $D/\tau_{\mathrm{L}}$ from above as $M \to 1$. This behavior is not surprising: as $M$ becomes shorter, the four-point function (21) begins to "self-average" over the inhomogeneity in a manner analogous to diffusion. Nevertheless, $D < v_{\mathrm{B}}^2 \tau_{\mathrm{L}}$ holds as a strict inequality in the inhomogeneous systems that we have studied numerically, even when $M = 1$ (no correlations among $D(x)$). Our violation of the relation $D = v_{\mathrm{B}}^2 \tau_{\mathrm{L}}$ is not limited to the regime of long range disorder.

So far, the discrepancies between $D$ and $v_{\mathrm{B}}^2 \tau_{\mathrm{L}}$ are only on the order of a few percent in our numerical data. Yet (29) implies that there is no possible upper bound on diffusion due to quantum chaos. Defining a natural probability measure on $D(x)$ as

$$\mathrm{p}(X)\mathrm{d}X \equiv \mathbb{E}[\Theta(X + \mathrm{d}X - D(x))\Theta(D(x) - X)], \tag{30}$$

we estimate that if

$$\mathrm{p}(X \to 0) \sim X^a, \quad \text{with } -\frac{1}{2} < a \leqslant 0, \tag{31}$$

then $v_{\mathrm{B}} > 0$ but $D = 0$.[5] We have looked for this parametric breakdown of the relationship between $D$ and $v_{\mathrm{B}}^2$ in smaller chains with $M = 1$. As shown in Figure 2, we see qualitative agreement (but quantitative disagreement) with our hydrodynamic predictions for $D$ and $v_{\mathrm{B}}^2 \tau_{\mathrm{L}}$ (accounting for finite size effects). As $a \lesssim 0$, we observe that the ratio $D/v_{\mathrm{B}}^2 \tau_{\mathrm{L}}$ becomes

---

[4]We must also normalize the value of $v_{\mathrm{B}}^{-1}$ found numerically by a factor very close to unity, to account for finite size effects. This factor depends only on $L$.

[5]Since our analytic calculation of $v_{\mathrm{B}}$ requires (24), and disorder is correlated over $M$ sites, we estimate that in a chain of length $L$ the minimal value of $D_0$ scales as $D_{\min} \sim (M/L)^{1/(1+a)}$. Requiring that $D_{\min}M^2 \gg 2\pi/\beta$ requires that $M^{3+2a} \gg L$, up to some dimensionless constant. Hence, we conclude that the inhomogeneous SYK chain with $D = 0$ but $v_{\mathrm{B}}$ finite only strictly exists in a somewhat subtle thermodynamic limit with $M$ and $L$ taken to $\infty$ simultaneously, making sure to obey $M^{3+2a} \gg L$.

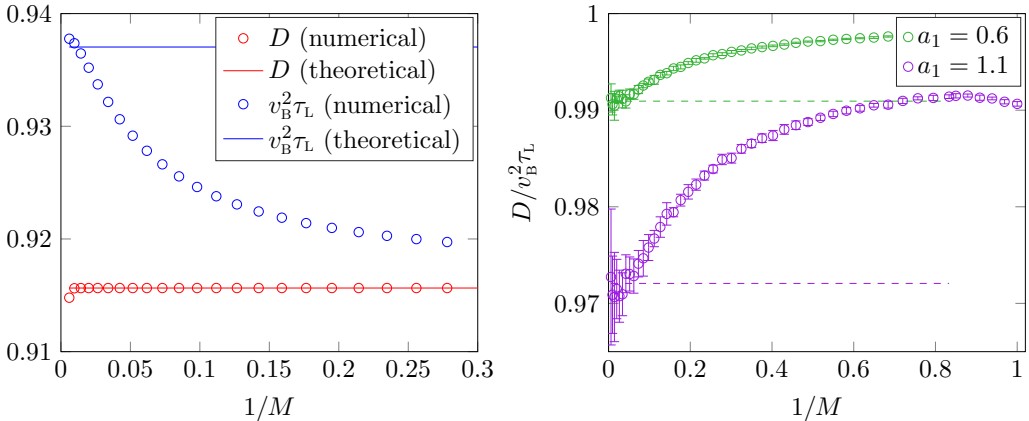

FIGURE 1: Left: the value of $D$ and $v_B^2 \tau_L$, predicted theoretically (from (20) and (26)) and found numerically, in an inhomogeneous chain with $J_1^2 = (1 + \mathscr{J}^2)^{-1}$, with $\mathscr{J} = a_0 + a_1 \cos(2\pi x/M)$ for $a_0 = 0.5$ and $a_1 = 0.6$. The trend of $v_B^2 \tau_L$ to decrease at smaller $M$ towards the "limit" set by diffusion is evident. Right: disordered profiles with $\mathscr{J} = a_0 + a_1 X$, with $X = \sum c_n \cos(\phi_n + k_n x)$ and $c_n$ and $\phi_n$ random variables chosen so that $X \sim O(1)$. We take $a_0 = 0.5$ and vary $a_1$, and study the ratio $D/v_B^2 \tau_L$ as a function of $M$ for different realizations of disorder. Discrepancies between the two are enhanced as $M$ becomes large, as expected. We have set $L = 6000$ and $\beta J = 25$.

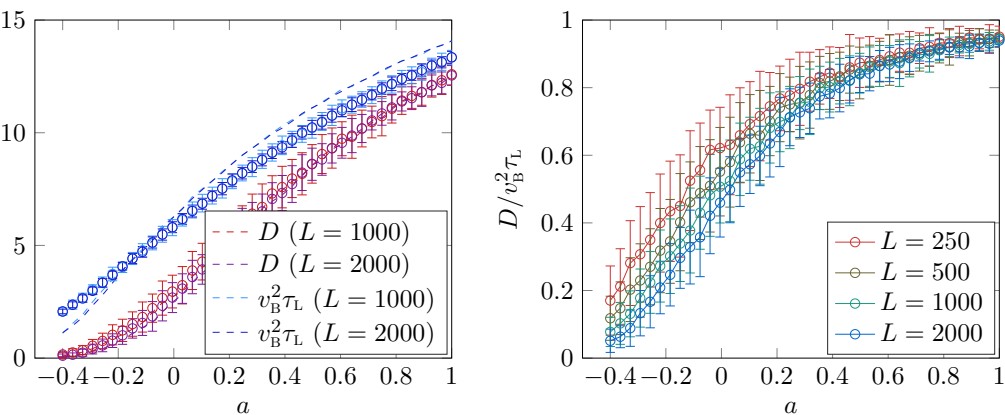

FIGURE 2: Left: Comparison of $D$ and $v_B^2 \tau_L$ as a function of $a$, for chains where $J_{1,x}^2$ are i.i.d. random variables drawn from the distribution $P(J_{1,x}^2 < X) = X^{1+a}\Theta(X)$. The circular data points with error bars are numerical data, and dashed lines are the mean of the hydrodynamic predictions for each chain. Qualitative agreement is observed. Right: the numerically computed ratio $D/v_B^2 \tau_L$ as a function of $a$. We see that this ratio rapidly drops for $a \lesssim 0$, as finite size effects become appreciable. This data is taken at $\beta J = 25$.

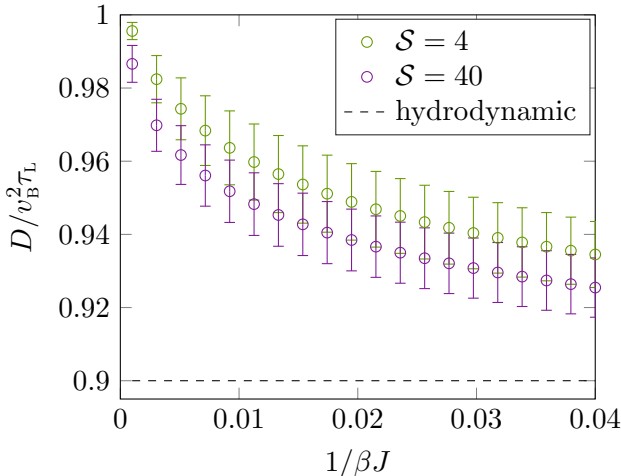

FIGURE 3: The temperature dependence of $D/v_{\text{B}}^2\tau_{\text{L}}$ for 'disordered' periodic lattices, where the 'hydrodynamic' prediction $D/v_{\text{B}}^2\tau_{\text{L}} \approx 0.9$. In the periodic lattice, $D(x)$ consists of a sum of $\mathcal{S}$ sine waves, which are tiled over a period $\mathcal{S} \times M$: $\mathcal{J} = |a + \frac{b}{\sqrt{\mathcal{S}}}\sum c_n \cos(\frac{2\pi n x}{M\mathcal{S}} + \phi_n)|$, with $b = 4a$ and $c_i$ and $\phi_i$ randomly chosen, given the constraint $D/v_{\text{B}}^2\tau_{\text{L}} \approx 0.9$. Increasing $\mathcal{S}$ decreases the ratio $D/v_{\text{B}}^2\tau_{\text{L}}$. This data is taken at $L = 4000$ and $M = 10$.

dependent on the length of the chain, and decreases for the longer chain. This provides evidence that even when $M = 1$, this inhomogeneous SYK chain may have $D = 0$ but $v_{\text{B}} > 0$ in the thermodynamic limit.

Finally, let us comment on the low temperature limit $\beta \to \infty$ (while, of course, taking $N \to \infty$ as well such that $\beta J \ll N$). At small enough temperature, keeping the inhomogeneity fixed, (24) will break down. Figure 2 suggests that the large $M$ limit is not required to obtain substantial deviations from $D = v_{\text{B}}^2\tau_{\text{L}}$. A more interesting subtlety that arises at very low temperature is the difference between periodic inhomogeneity and random inhomogeneity. For periodic inhomogeneity, one can diagonalize the matrix $C_{xy}$ as a periodic tight-binding hopping matrix in a larger unit cell, and at low enough temperatures, the solution to (23) can be well-approximated by considering only the physics of the lowest band. In this regime, one will recover $D = v_{\text{B}}^2\tau_{\text{L}}$. As the period of the periodic inhomogeneity grows longer, the temperature above which $D = v_{\text{B}}^2\tau_{\text{L}}$ decreases. We expect that for random inhomogeneity (where the eigenstates do not form bands, but are in fact localized) one finds $D < v_{\text{B}}^2\tau_{\text{L}}$ at all finite temperatures, and provide some numerical evidence for this in Figure 3.

## 4 Outlook

We have presented a modification of the SYK chain model of [34], in which there is an upper bound on the diffusion constant: $D \lesssim v_{\text{B}}^2\tau_{\text{L}}$. As we pointed out in the introduction, this suggests that there is no (simple) generic bound relating transport and quantum chaos in all strange metals.

One might ask whether our violation of (2) could be found in a "homogeneous" model that does not rely on explicit translation symmetry breaking in the low energy effective description. In the SYK chains that we have studied, this is easy to accomplish at leading order in $1/N$, because there is no difference betwen averages over annealed disorder vs. quenched disorder.

Hence, consider the partition function

$$Z = \int DJ_{1,x}^2 \, \mathbb{P}[J_{1,x}^2] Z_{\text{SYK}}[J_{1,x}^2], \tag{32}$$

with $Z_{\text{SYK}}$ the partition function defined in (6), and $\mathbb{P}[J_{1,x}^2]$ a translation invariant function where $J_{1,x}^2$ have support on some finite domain. We may think of $\mathbb{P}[J_{1,x}^2]$ as either a partition function for the "slow" dynamical variables $J_{1,x}^2$, or as accounting for certain correlated non-Gaussian fluctuations in the random variables $J_{ijkl,x}$ and $J_{ijkl,x}'$ of the microscopic Hamiltonian (4).[6] So long as $D \leqslant v_{\text{B}}^2 \tau_{\text{L}}$ for each choice of $J_{1,x}^2$, by linearity, this inequality will remain true even in the homogeneous model (32) after averaging over the ensemble $\mathbb{P}[J_{1,x}^2]$ of random couplings.

## Acknowledgements

We thank Mike Blake for helpful comments on a draft of this paper. AL was supported by the Gordon and Betty Moore Foundation's EPiQS Initiative through Grant GBMF4302. YG and XLQ are supported by the National Science Foundation through the grant No. DMR-1151786, and by the David & Lucile Packard Foundation.

## A  Derivation of Effective Action

It is convenient to expand about the saddle using renormalized variables $g_x, \sigma_x$ defined by

$$G_x(\tau_1, \tau_2) = G^{\text{s}}(\tau_1, \tau_2) + |G^{\text{s}}(\tau_1, \tau_2)|^{-1} g_x(\tau_1, \tau_2),$$
$$\Sigma_x(\tau_1, \tau_2) = \Sigma^{\text{s}}(\tau_1, \tau_2) + |G^{\text{s}}(\tau_1, \tau_2)| \sigma_x(\tau_1, \tau_2), \tag{33}$$

where we have rescaled the fluctuation fields $g_x, \sigma_x$ by prefactors $|G^{\text{s}}|^{-1}$ and $|G^{\text{s}}|$. It should be noticed that although the saddle point is uniform in space and translation invariant in time, the fluctuation fields have generic space-time dependence.

Now we expand the effective action to second order in the fluctuation fields $g, \sigma$, which leads to

$$S_{\text{eff.}}[g, \sigma] \approx S_{\text{eff}}^{\text{s}} - \frac{1}{4} \int d^4\tau \sum_x \sigma_x(\tau_1, \tau_2) G^{\text{s}}(\tau_{13}) \cdot |G^{\text{s}}(\tau_{34})| \cdot G^{\text{s}}(\tau_{42}) \cdot |G^{\text{s}}(\tau_{21})| \sigma_x(\tau_3, \tau_4)$$
$$+ \int d^2\tau \left( \sum_x \frac{1}{2} \sigma_x(\tau_1, \tau_2) g_x(\tau_1, \tau_2) - \frac{3J^2}{4} \sum_{x,y} g_x(\tau_1, \tau_2) S_{xy} g_y(\tau_1, \tau_2) \right). \tag{34}$$

The spatial kernel $S_{xy}$ is a tight-binding hopping matrix

$$S_{xy} = \delta_{xy} + \frac{1}{3J^2} \left[ (-J_{1,x}^2 - J_{1,x-1}^2) \delta_{xy} + \delta_{x,y-1} J_{1,x}^2 + \delta_{x,y+1} J_{1,x-1}^2 \right] = \delta_{xy} - C_{xy},, \tag{35}$$

---

[6]In this latter case, $J_{ijkl,x}$ and $J_{ijkl,x}'$ are no longer completely independent random variables. The reason for this is that after integrating over $\mathbb{P}[J_{1,x}]$ on a single site, we generically find $\int dJ_1 \mathbb{P}(J_1) \exp[-N^3 \sum J_{ijkl}'^2 / 2J_1^2] \neq \prod_{ijkl} \mathscr{F}(J_{ijkl}')$. Since the joint probability distribution of the $J_{ijkl}'$ does not factorize, we conclude that these random variables are no longer independent. Hence, integrating over fluctuations in $\mathbb{P}[J_1]$ restores translation invariance, but necessarily introduces non-trivial correlations between the $J_{ijkl,x}$ and $J_{ijkl,x}'$.

with $C_{xy}$ defined in (13). Next we integrate out $\sigma_x$ and obtain a quadratic action for $g_x$ alone. We define $\widetilde{K}$ as the (symmetrized) four-point function kernel of the SYK model:

$$\widetilde{K}\left(\tau_1, \tau_2; \tau_3, \tau_4\right) = 3J^2 G^s(\tau_{13}) \cdot |G^s(\tau_{34})| \cdot G^s(\tau_{42}) \cdot |G^s(\tau_{21})|. \tag{36}$$

We have defined $\tau_{ij} \equiv \tau_i - \tau_j$. The effective action of $g_x$ is therefore

$$\begin{aligned} S_{\text{eff.}}[g] = S_{\text{eff}}^s + \frac{3J^2}{4} \int \mathrm{d}^4\tau \sum_{x,y} g_x(\tau_1, \tau_2) \times \\ \left[\widetilde{K}^{-1}\left(\tau_1, \tau_2; \tau_3, \tau_4\right)\delta_{xy} - S_{xy}\delta(\tau_{13})\delta(\tau_{24})\right] g_y(\tau_3, \tau_4), \end{aligned} \tag{37}$$

The leading contribution comes from the soft modes which can be identified as induced by reparametrization $f_x \in \text{Diff}(S^1)$. They can be interpreted as energy fluctuations. We can linearize these modes and write them in Fourier space: $f_x(\tau) = \tau + \epsilon_x(\tau)$, $\epsilon_n = \int_0^\beta \mathrm{d}\tau e^{\frac{2\pi i n \tau}{\beta}} \epsilon(\tau)$. For such modes, we know from [32] that the kernel obeys

$$\left(\widetilde{K}^{-1} - 1\right)\epsilon_{n,x} = \frac{\alpha|n|}{\beta J}\epsilon_{n,x} \tag{38}$$

Following [32], we now find the following effective action for the linearized modes given in (12). The additional numerical prefactors in (12), including $|n|(n^2 - 1)$, come from properly normalizing the eigenvector $\epsilon_{n,x}$.

## B Eigenvalues of Inhomogeneous Diffusion Matrix

The following argument is reminiscent of [10]. Let us define the matrix

$$Q_{xy} = \delta_{xy} e^{iqx}, \tag{39}$$

and consider the limit $q \to 0$. We postulate that the lowest lying eigenvalues of $\widetilde{C}_{xy}$ are proportional to $D_{\text{eff}}q^2$:

$$\widetilde{C}_{xy}u_y = D_{\text{eff}}q^2 u_x. \tag{40}$$

Multiplying on both sides by $Q$ we obtain

$$Q_{xz}(D^\mathsf{T} S D Q^{-1})_{zw}(Qu)_{wy} = D_{\text{eff}}q^2(Qu)_x. \tag{41}$$

We now look for a series solution to this equation of the form

$$Qu = u_0 + iqu_1 - q^2 u_2 + \cdots. \tag{42}$$

Such a series expansion is reasonable – we expect in any finite chain that the lowest few eigenvectors are delocalized, and have confirmed this numerically. At $O(q^0)$, the $Q$ matrix is simply the identity, and so we must take

$$(u_0)_x = 1. \tag{43}$$

At $O(q^1)$, we find the equation

$$Q_{xz}\left[(D^\mathsf{T} S D Q^{-1})_{zw}(u_0 + iqu_1)_w\right] = 0. \tag{44}$$

$Q$ is invertible, and hence we conclude that if $u_1$ is non-trivial, the left-most $D$ must act on a non-trivial vector. Because $D$ has a one-dimensional kernel we conclude

$$c(u_0)_z = (SDQ^{-1})_{zw}(u_0 + iqu_1)_w \tag{45}$$

We may fix the constant $c$ as follows. First left-multiply by $S^{-1}$, and keep only first order terms in $q$, to obtain

$$cS^{-1}_{zw}(u_0)_w = (DQ^{-1})_{zw}(u_0)_w + \mathrm{i}qD_{zw}(u_1)_w \approx -\mathrm{i}qQ^{-1}_{zw}(u_0)_w + \mathrm{i}qD_{zw}(u_1)_w. \tag{46}$$

In the second equality, we have exploited the fact that $Q^{-1}u_0$ is an eigenvector of $D$ of eigenvalue $1 - \mathrm{e}^{\mathrm{i}q}$. Now, we perform a non-rigorous sleight-of-hand: at leading order in $q$, we may treat $qQ^{-1}_{zw}(u_0)_w = q(u_0)_w$. We may then left-multiply by $u_0$, to remove only the second term of (46), and obtain

$$c = -\mathrm{i}q\frac{u_0 \cdot u_0}{u_0 \cdot S^{-1}u_0}. \tag{47}$$

On physical grounds, this is the statement that the eigenvector looks a lot like a plane wave, and this seems to be true numerically. We now have the $\mathrm{O}(q)$ corrections to the eigenvector $u_y$, and to leading order $q$ use the variational principle to 'exactly' compute the effective diffusion constant. We find

$$(u_0 + \mathrm{i}qu_1) \cdot D^{\mathsf{T}}SD \cdot (u_0 + \mathrm{i}qu_1) = q^2c^2u_0 \cdot S^{-1}u_0 = D_{\mathrm{eff}}q^2u_0 \cdot u_0 \tag{48}$$

which gives us, for a periodic chain of $L$ sites:

$$\frac{1}{D_{\mathrm{eff}}} = \frac{1}{L}\sum\frac{1}{D_x}. \tag{49}$$

One way to rigorously obtain (46) is to extend a chain of length $L$ into an infinite chain by tiling the same $S_x$ repeatedly: $S_{x+L} = S_x$. Using the discrete translation symmetry and choosing the appropriate definition of $q$, one can demand $u_x = u_{x+L}$, and then sum over only $L$ sites in (46), killing the right-most term. In practice, we have often found this tiling to be unnecessary numerically.

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
