# Peer review of "Energy diffusion and the butterfly effect in inhomogeneous Sachdev-Ye-Kitaev chains"

_SciPost Physics, doi:SciPost Phys. 2, 018 (2017)_

## Round 1 · Referee Report · Moshe Rozali · 2017-4-4

Strengths
The relation between chaos and transport in quantum many body systems is an interesting and important subject. The manuscript is concerned with the status of a particular conjectured relation between the two. That conjecture already has some counter-examples, and might be puzzling because the measure of chaos they use has to do with short times whereas the diffusion probes the system's dynamics on longer time scales. Nevertheless, it is clarifying to concretely calculate the conjectured quantities in some specific examples and discuss the putative relation between them. This short paper does that, it is clear and well-written and I recommend publication.
Weaknesses
n/a
Report
n/a
Requested changes
none.

---

## Round 1 · Referee Report · Anonymous · 2017-4-24

Strengths
1) The paper considers the interesting question of a possible relation between diffusion and chaos in the context of SYK chains. Even though this has been asked before in the context of SYK, the authors generalise previous results to account for site dependent statistics for the random coupling constants of the model.
2) In this generalised setup, the authors derive the effective action (7) for the collective bilocal field. This can be useful for other researchers working on similar topics.
Weaknesses
1) It would be beneficial for the reader to see the logic behind deriving the effective action (10) for the reparametrisation modes. This is a central result which the authors use in order to discuss diffusion in the given model. It is true that the paper is very close to [33] but more details would be useful for the reader.
Report
The authors consider the question of a possible connection between chaos and diffusion in the context of a model which is a generalisation of the SYK model. I find the question and the conclusions of the paper interesting. This is an interesting paper which is certainly worth considering for publication.
Requested changes
1) -

---

## Round 2 · Author Response

We thank the referees for their positive feedback. We have edited the manuscript along the lines suggested -- see the list of changes. We hope that these changes satisfy the referees and the paper can be published in its present form.

---

## Round 2 · List of Changes

1) We have added Reference 12 in the introduction as an example of a violation of a postulated relation between charge diffusion and v_B.

2) We have modified the main text in Section 2 around Equations 7-11, along with adding a new Appendix, in order to address a referee's concern that our derivation of the effective action should be more self-contained.

---

## Editorial Decision

published